# Secured and Deterministic Closed-Loop IoT System Architecture for Sensor and Actuator Networks

**DOI:** 10.3390/s22103843

**Published:** 2022-05-19

**Authors:** Hyeon-Su Kim, Yu-Jin Park, Soon-Ju Kang

**Affiliations:** 1School of Electronics Engineering, College of IT Engineering, Kyungpook National University, 80 Daehakro, Bukgu, Daegu 41566, Korea; beeeaaar@knu.ac.kr; 2Center of Self-Organizing Software, Kyungpook National University, 80 Daehakro, Bukgu, Daegu 41566, Korea; ilbsyjp@knu.ac.kr

**Keywords:** closed-loop, IoT environment, sensor-actuator network

## Abstract

Sensors, actuators, and wireless communication technologies have developed significantly. Consequently, closed-loop systems that can be monitored and controlled by devices in IoT environments, such as farms and factories, have emerged. Such systems are realized by means of cloud-level and edge-level implementations. Among them, with a model that generates real-time control decisions at the cloud level, it might be difficult to ensure the determinism of real-time control due to communication overheads. In addition, if the actuator is remotely controlled at the cloud level, it is difficult to secure control safety against external hacking or device malfunction. Herein, we propose a system architecture that can fulfil real-time performance and safety requirements with two commonly used devices, Field Edge Unit (FEU) and Current Sensing Tag (CST), in a closed-loop IoT environment. By using these devices, we designed a special architecture that can be commonly used in various closed-loop sensing and actuating applications. In this study, the proposed architecture is evaluated by applying it to a smart farm application.

## 1. Introduction

Owing to the development of sensors, actuators, and wireless communication technologies, the adoption of closed-loop systems in IoT environments, such as farms and factories, has increased [1,2,3]. These systems are largely divided into cloud-level and edge-level systems, and the use of cloud-level closed systems is increasing owing to the convenience offered by cloud-level IoT platforms [4,5,6]. However, applications such as the ones that involve determining actuator operation based on sensor values require frequent interaction with edge devices. When running such an application on a cloud server, the aforementioned frequent interactions increase communication overhead with the cloud. This overhead makes it difficult for the system to satisfy the system requirements and furthermore to guarantee determinism. If remote control through the cloud is adopted, a possibility of system malfunction emerges, although such damage is non-fatal. For instance, this can occur when an outsider directly operates the actuator from a close distance or the content of remote communication is leaked. Owing to these weaknesses, remote control hinders the safety of system operation.

In this study, we propose a system architecture to solve the above problems. The proposed architecture can control the environment in an edge-level closed-loop manner, including wired sensor and wired and wireless actuator control. In addition, the current consumption of wireless actuators is measured to check whether control actions have been executed normally. The architecture is largely divided into the Field Edge Unit (FEU), current sensing tag (CST), and relay hub, and their roles are as follows: The FEU is implemented to operate closed-loop systems at the edge level. In place of the existing closed-loop system at the cloud level, our architecture was implemented and used to determine actuator behavior based on sensor values at the edge level. CST is used in combination with the aforementioned remotely controllable AC socket, and it reads the current through the AC line inside the socket to the sensor. Based on these data, CST sends a message when the operation of the actuator connected to the socket changes. In the architecture proposed herein, CST is adopted to remotely use several actuators plugged into 220V sockets and detect changes in actuator operation. The relay hub is implemented and used as for relaying between the edge-level devices (FEU, CST) and cloud servers. The relay hub is connected to the cloud through ethernet and to edge devices through Bluetooth Low Energy (BLE). In addition, the relay hub sends a signal to control the remote actuator when requested by the FEU. Furthermore, the relay hub combines changes in the FEU sensor data, actuator operation requests, and current consumption of actuators obtained through CST to determine and inform the user of abnormal situations. There are four situations to be considered: (1) When current consumption is not detected after the remote actuator control command is processed. (2) A current consumption is detected in the absence of any command. (3) The sensor value does not change, even though the actuator is operated according to the system command. (4) There is no system command, and the actuator is not working, but the sensor value changes significantly. As mentioned earlier, ensuring the determinism of closed-loop behavior and securing safety in remote control are key goals of the proposed architecture [7]. The results of a performance evaluation confirmed that the proposed architecture is superior in terms of the determinism of operation than the closed-loop model at the cloud level, and it can accurately recognize and notify users about abnormal situations during remote control.

The remainder of this thesis is organized as follows: Section 2 provides an overview of the existing studies. In Section 3, design requirements, overall system structure, and system sequence are described. Section 4 provides details such as the implementation of two the subsystems, namely FEU and CST. The performance evaluation conducted in this study is presented in Section 5. Finally, a summary of this study and our concluding remarks are given in Section 6.

## 2. Related Works

Various IoT technologies, communication methods, and decision-making algorithms are used in the smart farm field. H. Navarro-Hellín et al. [8] used a wireless sensor node for the irrigation water management system, and transmitted sensor data to the server through the cellular network of GSM/GPRs. In order to implement a large-scale smart farm, a highly scalable wireless communication network is required, and a cellular network with a wide operating range is mainly used. In this study, we propose a platform that controls a group of sensor nodes within a smaller area and adjusting water supply. In the proposed platform, edge-level close-loop control is performed; however, there is a plan to use the existing WiFi or cellular network to store sensor data to the server through the gateway device in future works.

Viani et al. proposed a fuzzy-based decision support system for irrigation management in a vineyard [9]. In the proposed system, they designed a wireless sensor and actuator network (WSAN) architecture to collect data and perform irrigation control. The actuator on/off was scheduled through the collected data and a decision support system based on fuzzy logic, effectively reducing water consumption of crops. The control platform in the smart farm must consider various environmental variables and control factors such as moisture control, nutrient supply, and disease prevention [10]. However, in this study, only the method of maintaining the soil with constant humidity through moisture control is considered.

In recent years, research has been actively conducted on the use of machine learning to make decisions for growing crops [11]. We are also considering applying server or TinyML-based edge-level machine learning technology to control based on collected sensor data in next study. Short-term moisture control requires an immediate response, so it is advantageous to handle it within the WSN network. However, in the case of long-term moisture control, various factors must be considered, so we plan to study a machine learning-based control platform by accumulating sensor data and result values on the server.

To solve the problem of closed-loop determinism at the cloud level, W. Lee et al. [12] proposed a model in which gateway nodes are placed between the cloud and edge levels. The model assumed a closed-loop system with multiple nodes and implemented a fog computing network by placing a computing node serving as a gateway between the cloud and edge levels to reduce cloud communication in a scenario where multiple nodes require communication. In this manner, they reduced direct communication between nodes and cloud servers and ensured determinism of the closed-loop system by using gateway computing nodes that possess more computing power than the edge-level nodes.

However, the above methods are either extremely expensive from the viewpoint of configuring a network and communication topology, or they cannot handle security problems. Herein, we propose a system architecture that can solve the abovementioned problems. The proposed architecture is implemented at the edge level, which reduces the communication overhead, despite frequent interactions with edge devices. Moreover, the proposed architecture can detect abnormal situations based on the current consumption of the remote-control actuator.

## 3. Materials and Methods

### 3.1. Application Example of Closed-Loop IoT Control

Herein, we design a remote-controlled environment system that remotely monitors plant conditions and supplies water or liquid fertilizers depending on the conditions, which is a representative example of closed-loop IoT networks [13,14]. Figure 1 shows the design and a schematic diagram of the smart farm stick implemented in this study. The smart farm stick is largely composed of an FEU and a CST. The FEU reads sensor values and creates actuator control signals in a closed-loop fashion. If the actuator is wired to the FEU, the FEU directly controls the actuator; otherwise, the control signal is transmitted to the relay hub.

Figure 2 shows a commercial AC socket that can be controlled remotely and a CST mounted therein. The CST measures the current consumption when the actuator is operated based on remote signals and creates an interrupt message based on the current consumption. In addition, because a remotely controllable 220 V socket is used, the system can use a 220 V commercial actuator as a remotely controllable actuator.

In this study, temperature is controlled using an electric heater, and this electric heater is connected to a remotely controllable commercial AC socket. The CST is installed in this socket, meaning that it can measure the current consumption of the electric heater. Moreover, the FEU can be used to acquire temperature sensor data. By using these data, our system can determine whether remote control has been performed as intended. The design requirements, overall configuration of the proposed architecture, and system sequences are covered in the sections that follow.

### 3.2. Design Considerations

When a closed-loop system is implemented at the cloud level, communication delay and communication processing overhead make it difficult to ensure system requirement of the operation. Moreover, during remote control of the actuator, many systems are only checked indirectly based on the values output by linked sensors, and it is difficult to ensure operational safety. The proposed architecture focuses on these two problems, and its implementation requirements are as follows.

First, it is necessary to reduce communication overhead with the cloud.

Figure 3 schematically shows the closed-loop system at the cloud level (left) and the closed-loop system at the edge level (right). In case of the closed-loop system at the cloud level, as shown in the figure on the left, communication between the edge level and the cloud level is inevitably frequent owing to the characteristics of closed-loop systems, which warrant frequent interactions with devices. This processing method causes significant communication delay and communication processing overhead. By contrast, if the application is executed at the edge level, as shown in the picture on the right, the overhead is reduced compared with that in the previous case. For this reason, we attempted to control the environment with the closed-loop system at the edge level.

Second, when controlling a remote actuator, it is necessary to ensure the safety of remote control by distinguishing whether the actuator is being operated under a normal system command or a non-system factor. If actuator operation is caused by a non-system factor, the system should warn the user.

The four abnormal situations that may occur during remote control, which the proposed architecture intends to provide warnings about, are as follows: (1) When the system issues an actuator control command, but current consumption of the actuator is not detected. (2) There is no control command in the system, but actuator current consumption is detected. (3) Current consumption is normally detected, and system command is issued, but there is no change in the associated sensor value. (4) There is no system command or current consumption, but the sensor value increases abnormally.

The existing system, which indirectly checks whether the actuator operates based on the sensor value, is limited in terms of capturing the abovementioned situations. Unlike conventional systems, the proposed architecture aims to distinguish and warn users of the abovementioned situations based on the following three factors: whether the system has issued commands, whether sensor values changed, and whether the actuator has consumed current.

### 3.3. System Achitecture Overview

An overview of the proposed architecture is presented in Figure 4. Users receive data through MQTT broker servers located in the cloud, send trigger messages, and edge-level hubs relay data between the servers and the edge devices through MQTT and BLE communication. The FEU is in charge of generating actuator control signals by reading environmental sensor values, and the CST is in charge of directly checking whether the remote actuator is actually turned on based on its current consumption.

The hub, FEU, and CST are described as follows: The FEU is a closed-loop system that periodically reads sensor values in IoT environments and determines the control time of actuators based on the sensor values. All key operational processes of a closed-loop system consisting of sensing-judgment-control signal generation occur at the edge level, and only a few messages, such as occasional sensor data transmission and remote actuator control requests, are delivered to the relay hub.

The CST is installed in a 220 V AC socket, and power application/disconnection to the socket can be controlled through an MQTT message. The CST is directly connected to the AC line within the socket such that it can measure the current consumed by the actuator. The CST divides the current consumption by a certain unit and generates a real-time interrupt based on changes in the actuator’s operation and sends an event message to the hub.

The relay hub is connected to the FEU and CST through BLE communication and to the cloud broker server through Ethernet MQTT communication. The basic role of the relay hub is to relay messages, such as device settings that the user wants to send to edge devices, sensor values, and post-control results, sent from edge devices to the user. In addition, the relay hub detects the four aforementioned abnormal situations that may be encountered in the remote-control situation. A Raspberry Pi4 [13] is used to implement the relay hub, and AWS IoT Core [14] is used as the broker server.

### 3.4. System Sequence

The architecture proposed herein stores the three factors mentioned above: whether the system commands actuator operation, whether current consumption occurs, and changes in sensor values. These factors enable the system to recognize the four abnormal situations mentioned above.

Figure 5 is a diagram showing how FEU, CST, and remote actuators operate in conjunction, and at which stage they store three factors (dotted boxes correspond to these elements) to determine abnormal situations in remote control.

First, when sensor data are transmitted from the FEU, the relay hub calculates and stores how different this sensor data are from the previous sensor data and how different it is from the target sensor data range (change in sensor value). When a request for operation of the remote actuator (whether the system has an actuator operation command) is received from the FEU, the relay hub sends an operation message to the AC socket to which the actuator is connected via Ethernet MQTT.

If the operation request is normally transmitted, power is applied to the AC socket to operate the actuator, which increases the current consumption and transmits an event message (whether a consumed current occurs) from the CST to the hub, based on the current consumption. The hub generates warnings through the three flags received in this way.

The Table 1 summarizes the abnormal situations detected during remote control based on the three factors mentioned above. In the case of rows 1 and 2, the system command and the current consumption actually generated by the actuator do not match. In row 1, there is a command but the actuator does not operate; in row 2, the actuator operates, even though there is no system command. In these two cases, there may be a problem with remote control due to external factors.

In the cases of rows 3 and 4, system control and actual actuator operation are performed normally, but the changes in sensor values do not match the target values. In these cases, the actuator and sensor settings may be problematic.

## 4. Detailed Design Implementation

### 4.1. Field Edge Unit

#### 4.1.1. Hardware Design of Field Edge Unit

Figure 6 shows the actual FEU board, and Figure 7 presents a schematic design diagram of the FEU system. The main role of the sensing board is to periodically check sensor values through RTC interrupts for generating control signals for linked actuators. If the actuator is wired, the FEU controls it immediately, and if it is remote-controlled, the control signal is transmitted to the hub. This operation is performed in a closed-loop fashion and implemented by adjusting the actuator’s operating time within a specified operating cycle depending on the difference between the current value and the target value of the associated sensor. The sensing board was designed as a Cortex-M0+ family chip to perform relatively light roles such as hardware interrupt service routine (ISR); FreeRTOS [15], an embedded OS, was ported to the chip.

The main role of the communication board is BLE communication with the relay hub, which receives messages from users, such as setting entities from the hub, or sends sensor data and actuator control signals to the hub. To use the complex BLE communication stack, the communication board uses an M4 series chip that offers higher performance than the sensing board, and UBINOS [16], an embedded OS, is ported to the chip.

#### 4.1.2. Event Sequence for Remote Sensing

Figure 8 shows the event sequence of the FEU. The first frame represents an RTC routine that is performed regularly. The main role of the FEU is to periodically collect sensor data and operate actuators by following the closed-loop method to ensure that the target environment remains constant. If the associated actuator is not directly connected to the FEU, the operation request for remote actuator is sent to the hub for processing. In addition to operating automatically according to the closed-loop setting, when a user command is received, actuator operation and sensor data transmission can be performed immediately, and the second and third frames in the above figure correspond to these operations.

#### 4.1.3. Procedural Algorithm for Closed-loop Control

Figure 9 schematically shows the closed-loop sequence at the edge level, which is the core role of the FEU. In the closed-loop control scheme of the FEU, sensor data collection and actuator control depending on the sensor data occur within the unit. Information about closed-loop operation, such as which sensor and actuator will be linked, how the actuator operates depending on the sensor data, and the RTC cycle, are received from the user through the hub and stored. When the RTC routine operates at regular intervals, the unit sends the sensor data and the result of actuator operation to the hub.

Herein, the term actuation ratio is used. This term refers to the ratio of the point in time at which the actuator is operated within the sensor cycle to the duration of the entire sensor. If the sensing period is 250 ms, and the time at which the actuator is operated is 50 ms, the actuation ratio is 20%. This is illustrated in Figure 10. For example, when the actuator starts to operate below 30% and stops at 40%, the actuation ratio is 100%, (Tbasic) below soil moisture is 30% (Vcurrent), and actuation ratio is 0%, when it exceeds soil moisture 40%. Between 30% and 40%, at least, the actuation ratio should be the basic ratio (Pdefault), and the actuation ratio in the section depends on the ratio of the current sensor value within the target sensor range. As it approaches the target sensor value (e.g., 40%), the actuation ratio is the basic ratio Pdefault%.

In addition, although this is not the basic operation of the FEU, the user device connected to the Ethernet MQTT broker may read sensor data and determine whether to operate the actuator, as shown in Figure 11. However, during such an operation, BLE, Ethernet communication delay, and communication processing overhead are inserted between sensor data collection and actuator operation. This makes it impossible to maintain the determinism of the operation. For this reason, the FEU is based on the edge-level closed-loop introduced earlier, and in the Performance Evaluation section, the two modes are compared in terms of the determinism of closed-loop operation.

### 4.2. Current Sensing Tag

#### 4.2.1. Hardware Design of Current Sensing Tag

The CST was mounted within a 220 V AC socket (Figure 12) to measure the current consumed by a home appliance, and it aims to classify the behavior of the home appliance based on the consumed current. In this study, actuators were connected and used instead of home appliances. Figure 13 schematically shows the CST hardware. Cortex M4 series chips were used as the main chips of the CST. To asynchronously control functions such as BLE communication and ADC function, UBINOS, an embedded OS, was ported. The ACS-712 current sensor for current measurement was connected to a 220 V AC hotline within the AC socket to measure the current consumed by the home appliance connected to the socket.

There are three types of ACS-712 [17] current sensors (5 A, 20 A, 30 A), and the current sensor ACS712ELCTR-20A-T used herein is the 20 A model. This model was selected because home appliances usually consume more than 1100 W (5 A); for instance, several types of heaters such as irons, coffee pots, and electric stoves (more than 1500 W).

#### 4.2.2. Event Sequence for Remote Control

Figure 14 shows the event sequence of CST. CST aims not to accurately check the current consumed by the home appliance but to classify the operation of the home appliance based on the detected current. For classification purpose, CST measures the current in the standby state before operation of the home appliance and stores this value as a bias, which corresponds to the initializing frame in Figure 14. The general use process after initialization corresponds to the after initializing frame. After initialization, CST measures the current periodically. If the measured current consumption differs by more than a certain level from the previously measured current consumption, that is, if it is determined that the operation of the home appliance changes, a message is sent to the user.

#### 4.2.3. Classification of Appliance Operations

During current measurement in the CST, the current enters in the form of a sine wave that has AC characteristics, and CST extracts and uses the amplitude of the sine wave. However, the goal of CST is not to accurately measure the current consumption but to classify the behavior of home appliances based on the measured current. For this reason, the measurement bias is not set to zero, and if there exists a current that is always consumed even if it is not controlled, that current is set as the bias. The current consumption is first measured when the connected home appliance is not operated during the initialization process (Figure 15) (considering whether there is current consumption even in the standby state), and the value is set as the bias. The step of determining the bias is the GET BIAS LINE stage in Figure 15. The next step is the validation step to determine whether CST can distinguish between the measured bias and the current consumed by the minimum operation that the system aims to detect.

During the validation stage, for a few seconds, the user must maintain the operation that is expected to consume the least current among the operations of the home appliance that user aims to detect, while the system checks whether the operating current can be distinguished from the bias current. The role of CST is to measure the current consumption of connected home appliances based on the specified bias by following the above-described processes, classify them into certain units, and classify them at least by on/off and operation level according to the type of home appliance.

Figure 16 is an amplitude graph (right) measured based on the power consumption row data graph (left) and bias of the electric heater. Through the signal in the right figure extracted from the signal in the left figure, it can be confirmed that the power consumption varies depending on the on/off heater or the intensity of operation, and can be classified according to the operation of appliance. Tests for various home appliances are covered in the section Performance Evaluation.

## 5. Performance Evaluation

### 5.1. Closed-Loop Comparison between Cloud Level and Edge Level on FEU

In this section, we compare the edge and cloud-level closed-loops in terms of operational determinism on FEU. In both cases, the target soil humidity was 30–40%, actuation duty ratio was 100% at 30% humidity, and actuation (water pump was used) operation duty ratio (default rate) was 30% just before 40% humidity. Therefore, in the ideal scenario, the soil humidity is 40% after control. In the current system, the server upload time for which an operation can be sustained without problems in consideration of communication delay and communication processing delay is 1.5 s, and at the edge level, the sensor value is updated internally at intervals of 250 ms.

Before comparing the closed-loop, the actuation ratio was set to 100%, and the sensor value was read at intervals of 200 ms to measure the performance of the test environment, Figure 17. The average growth rate of soil moisture in the test environment was 0.5383712 [%/200 ms]. The total number of samples in Figure 18 is 50, of which the value was not updated only twice. Through this, it is judged that there will be no problem with the actual test sample rate (250 ms).

Figure 18 shows a graph of a cloud model. Among them, a signal that changes steeply is the actuation duty ratio, and the other signal indicates soil humidity. When soil moisture decreases to less than 30%, the actuator starts to operate and stops beyond the target of 40%; in the 30–40% section, the system controls the actuation ratio by following the formula given in Figure 18. Figure 19 shows the distribution when the cases were classified depending on how much additional water was provided after the target soil moisture of 40% was reached post 20 measurements, as illustrated in Figure 18. In the case of cloud implementation, owing to the long cycle of receiving data due to the communication overhead, the actuator was not controlled delicately, as shown in the graph, and the sensor value was exceeded by a considerable margin, as shown in Figure 18. In addition, it is not easy to predict the operation because the excess sensor value does not invariably exceed a certain error range.

Figure 20 and Figure 21 show the results obtained at the edge level in the same manner as those in Figure 18 and Figure 19. Unlike the previous case, the graph shows that the actuator was controlled more delicately, and the chart shows that the control results are closer to target value and more predictable. Based on the above results, it is difficult for the cloud model to ensure system requirement of the closed-loop operation. For this reason, the proposed architecture minimizes tasks in remote environments based on decision-making at the edge level.

### 5.2. Classification of Appliance‘s Operations on CST

The core function of CST is whether the operation of the home appliance can be distinguished by measuring the current consumption of the home appliance connected to the commercial AC socket. The tests used four home appliances: humidifiers, electric rice cookers, fans, and electric heaters; and the official power consumption indicated on the product is 200 W, 1000 W, 80 W, and 2000 W, respectively.

Figure 22 shows a graph of the extracted current consumption amplitudes of each of the four home appliances. For the humidifier and electric rice cooker, the power consumption levels of which exceed 200 W, on/off classification, the most basic operation classification, is easy. In the case of the electric heater, the current consumption level of which is high, there was adequate discernment to distinguish not only the off state but also a few operating modes. However, in case of the electric fan, the off state and motion with the largest current consumption were clearly distinguishable, but motions such as weak wind and mild wind were not discernible, as can be inferred from the center of the graph. As a result of the test, the higher the current consumption, the better the discernment ability, and on the contrary, it can be seen that there is a limit to the discernment between operations when power consumption is low, such as a fan. However, since the current test was performed at the 20 A version among ACS-712 sensors, it is expected that the discrimination will be improved if the 5 A version sensor among the same ACS-712 sensors is replaced when low current consumption should be targeted.

### 5.3. Abnormal Situations in Remote Actuator Control

One of the core functions of the architecture presented in the study is to recognize and inform the user of abnormal situations that may occur when remote control is performed. The situation of the test assumed that the air temperature should be controlled. In the test, an air temperature sensor was installed in the FEU, and an electric heater was plugged into a commercial socket where the CST was installed.

Figure 23 schematically shows how the response time was measured in the actuator off case, which is an abnormal situation, and the measurement results. In this case, measurement of the response time was started when the user sent an operation request to the hub. After this, a control signal was sent to the actuator, and the measurement ended when the user received a warning signal because no consumption current was detected. The time delay between departure of the control signal and arrival of the response was 507.35 ms on average. This is because if current consumption occurs in the actuator after the control signal is transmitted, it can be sent directly to the user in an event-triggered manner, but in this case, the system was meant to wait for 500 ms because current consumption did not occur.

Figure 24 shows the exception situation involving a malfunction, similar to Figure 24. This case is different from the previous case in which the system sends a control signal and waits for a current consumption message. The system did not send a control command, but as soon as the current consumption message arrived in an event-triggered fashion, a warning was generated. Therefore, unlike the previous case, the response time was short.

Figure 25 shows the abnormal situations, namely actuator range problem (left) and environment problem (right). Unlike the previous cases, these two cases require consideration of the third factor of inference, which refers to changes in sensor values. The system collects sensor values for a predetermined period and detects any abrupt changes in them. The reasoning process and the process of sending warning signals to the user are identical. For this reason, we grouped the two cases together to measure the response time. In addition, because the process of inferring any changes in the sensor values may vary depending on the user’s settings and the RTC period, in the test the response time was measured when a warning signal was sent to the user after an abnormal situation occurred, which refers to the communication delay between the hub and the user. Therefore, by comparing the above delay and the delay in the previous two cases, the difference between the system’s situation inference and the user’s notification can be observed.

## 6. Conclusions

In this study, we propose a system architecture that can compensate for the problems related to the determinism of closed-loop systems and the safety of remote control through the cloud. The FEU was implemented and used to ensure sensing and control of the closed-loop operation in the proposed architecture. In the case of remote control through the cloud, actuator current consumption was measured using the CST implemented herein. The proposed architecture ensured the safety of control by detecting and warning the user of abnormal situations that occurred during remote control based on the following three factors: whether the system provided control commands, changes in sensor values, and actuator current consumption. In the proposed architecture, we assumed a smart farm environment as the IoT control environment, and a test was conducted on medium-sized papaya pots. A comparison of closed-loop performance at the cloud and edge levels confirmed that the proposed architecture can ensure faster response time than closed-loop control at the cloud level. In the test results, about 65% of control errors occurred at the cloud level and about 20% in the edge level. Moreover, by measuring the response time in the four defined error situations, a low delay rate was observed in most situations. Simulation of four abnormal remote-control situations confirmed that all four cases were recognized normally, and a warning message was delivered to the user within the response time to ensure real-time control. In future studies, we aim to predict changes in environmental data by applying data analysis techniques, such as regression and deep learning [18] to the data uploaded to the cloud and improve the operational determinism of cloud-level closed-loop systems by constructing an algorithm that adjusts the control cycle depending on the current sensor value.

## Figures and Tables

**Figure 1 sensors-22-03843-f001:**
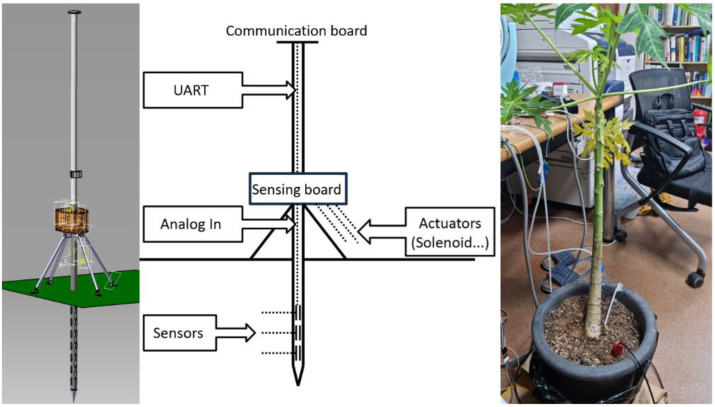
Smart farm stick and plant used in test.

**Figure 2 sensors-22-03843-f002:**
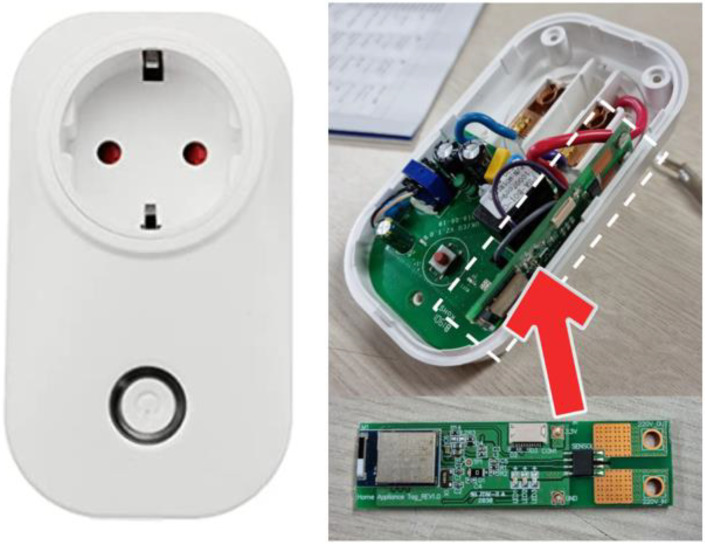
Commercial AC socket and current sensing tag.

**Figure 3 sensors-22-03843-f003:**
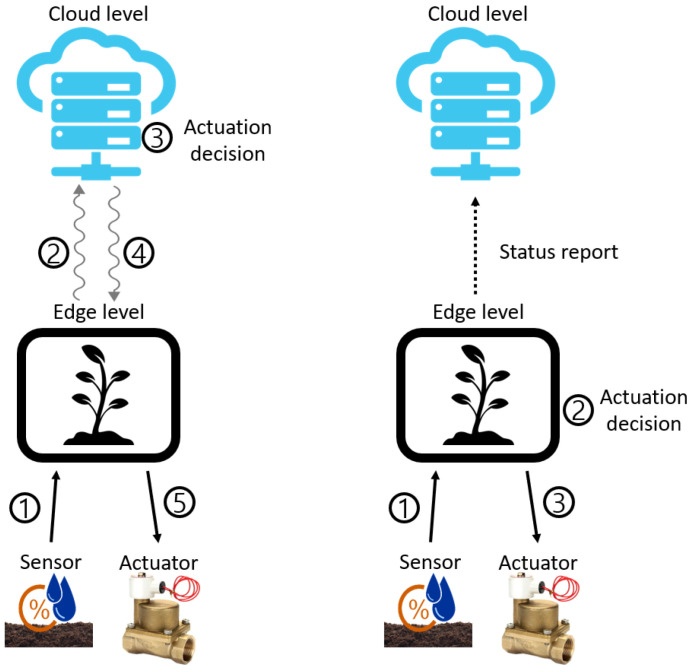
Closed-loop system at the cloud level vs. edge level.

**Figure 4 sensors-22-03843-f004:**
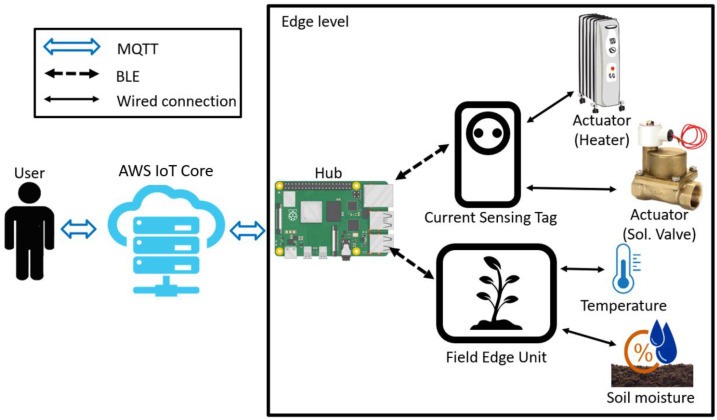
Closed-loop system at the cloud level vs. edge level.

**Figure 5 sensors-22-03843-f005:**
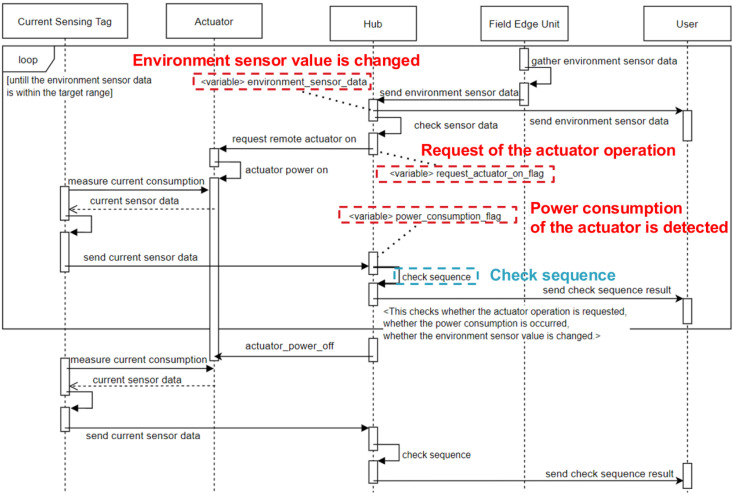
System sequence of proposed.

**Figure 6 sensors-22-03843-f006:**
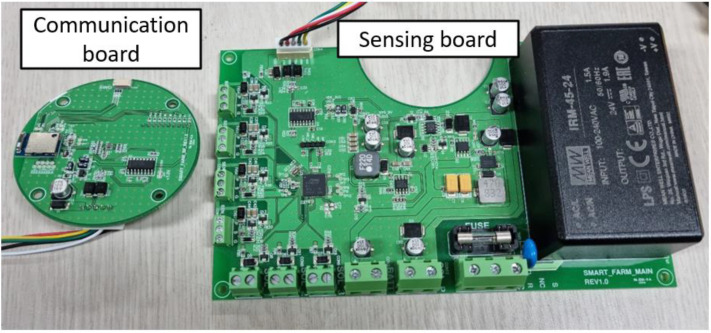
Field Edge Unit boards.

**Figure 7 sensors-22-03843-f007:**
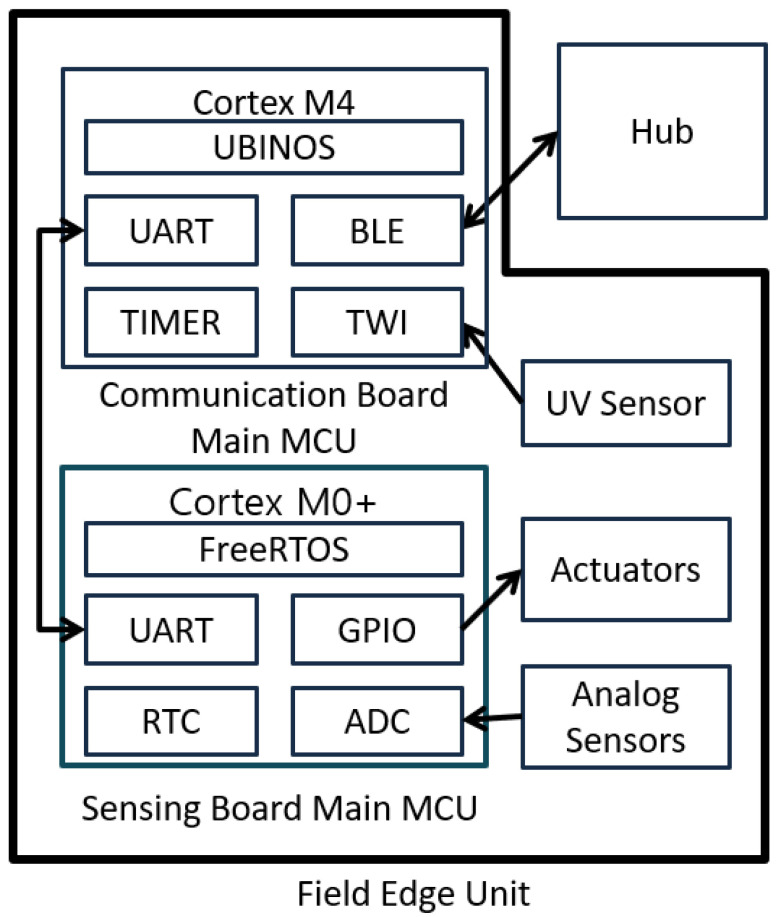
Hardware block diagram of Field Edge Unit.

**Figure 8 sensors-22-03843-f008:**
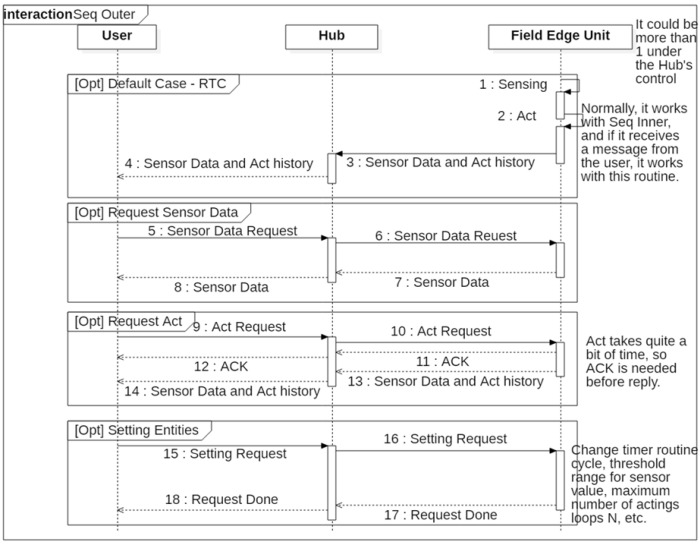
Event sequence of Field Edge Unit.

**Figure 9 sensors-22-03843-f009:**
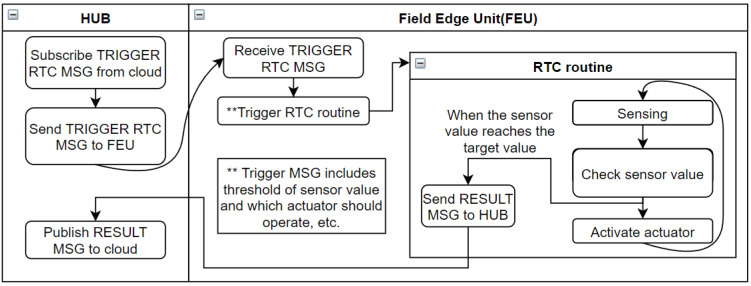
Schematic diagram of closed-loop control when the system determines its actuator’s action at the edge level.

**Figure 10 sensors-22-03843-f010:**
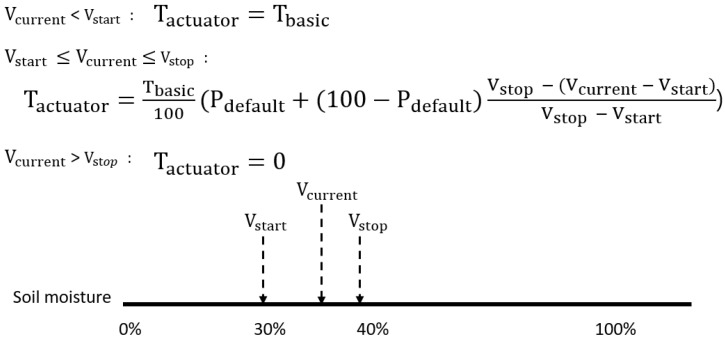
Actuation ratio depending on the target sensor value and the current sensor value.

**Figure 11 sensors-22-03843-f011:**
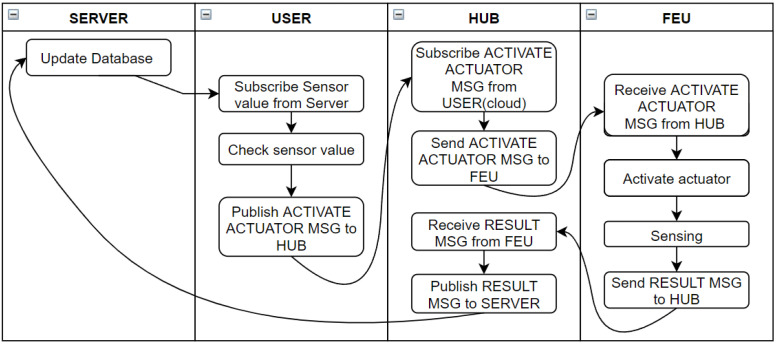
Schematic diagram of closed-loop control when the system determines actuator action at the cloud level.

**Figure 12 sensors-22-03843-f012:**
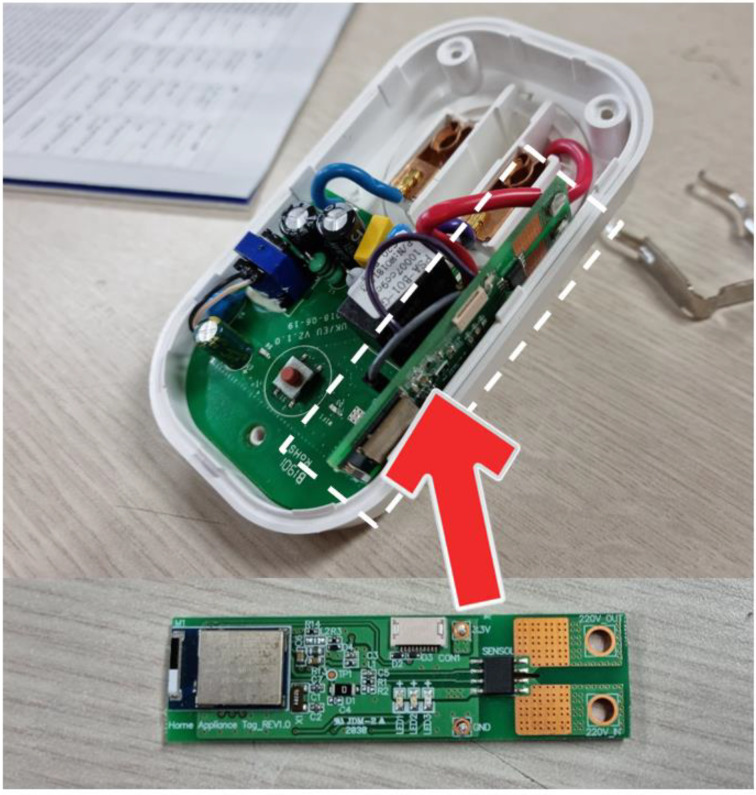
Current sensing tag board.

**Figure 13 sensors-22-03843-f013:**
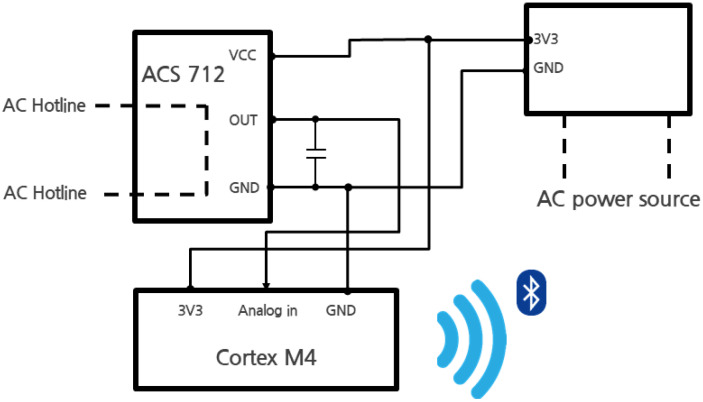
Hardware block diagram of Current Sensing Tag.

**Figure 14 sensors-22-03843-f014:**
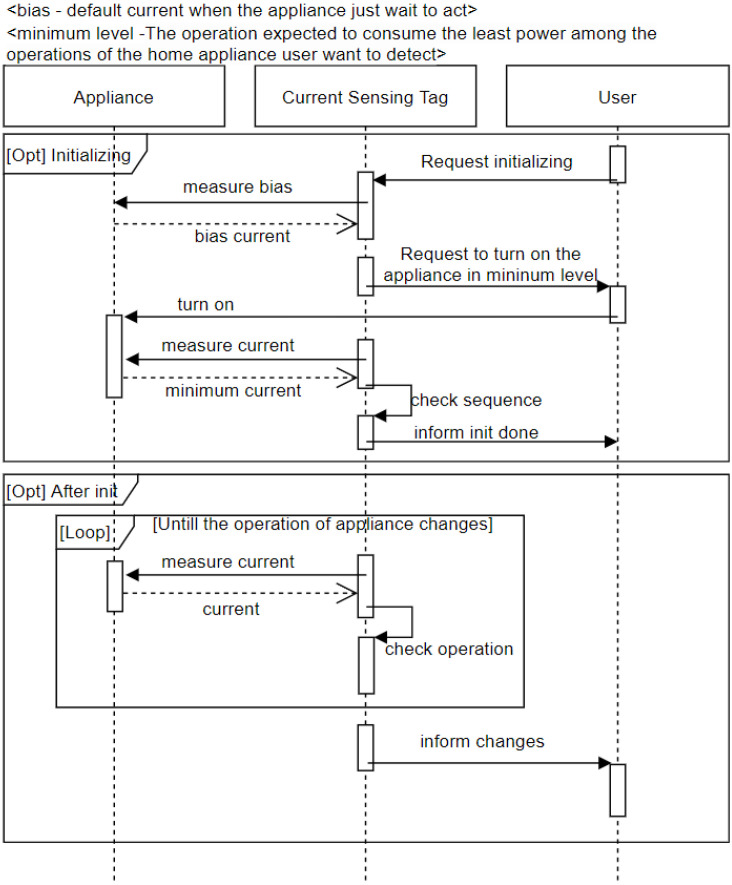
Event sequence of Current Sensing Tag.

**Figure 15 sensors-22-03843-f015:**
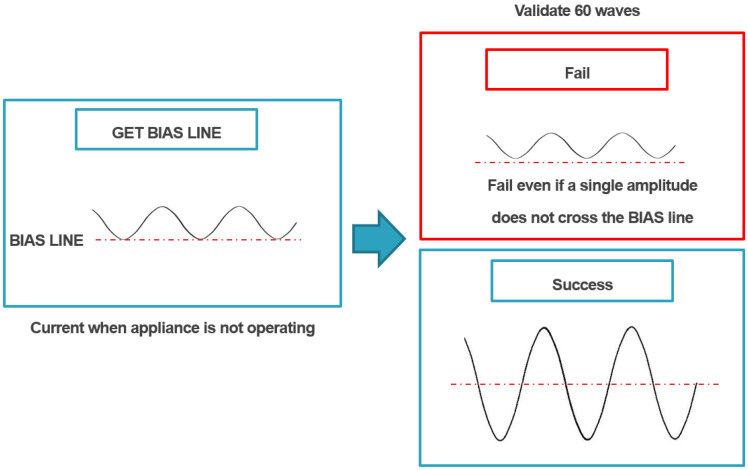
Initialization process.

**Figure 16 sensors-22-03843-f016:**
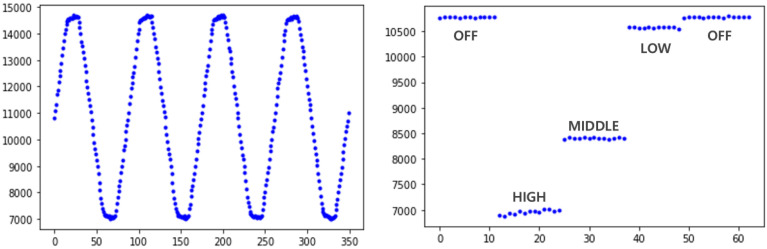
Current consumption of the heater.

**Figure 17 sensors-22-03843-f017:**
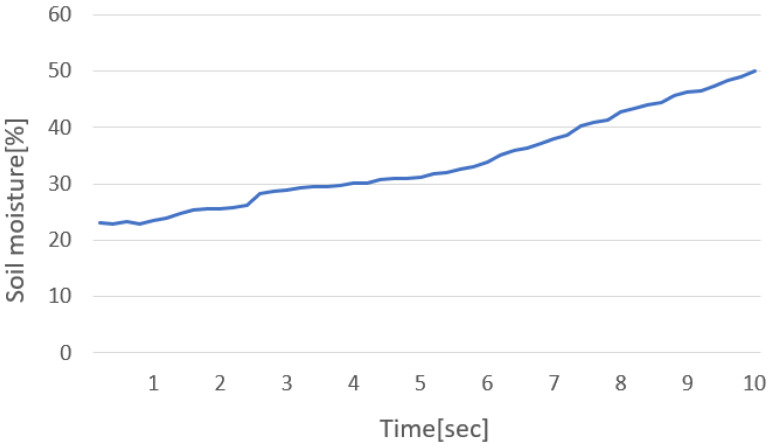
Current consumption of the heater.

**Figure 18 sensors-22-03843-f018:**
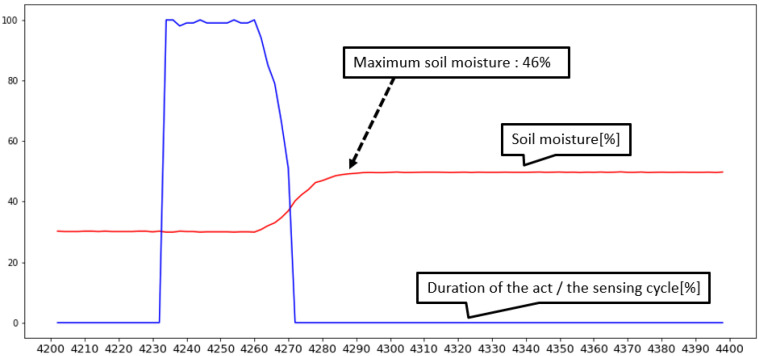
Soil moisture sensing value reflecting actuator’s action in cloud level closed-loop control.

**Figure 19 sensors-22-03843-f019:**
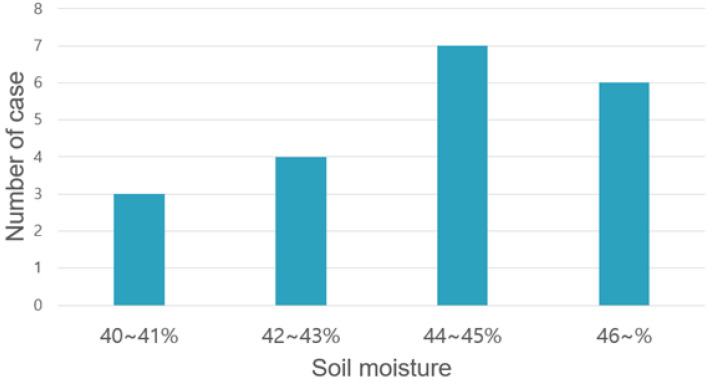
Case distribution according to max soil moisture in cloud level closed-loop control.

**Figure 20 sensors-22-03843-f020:**
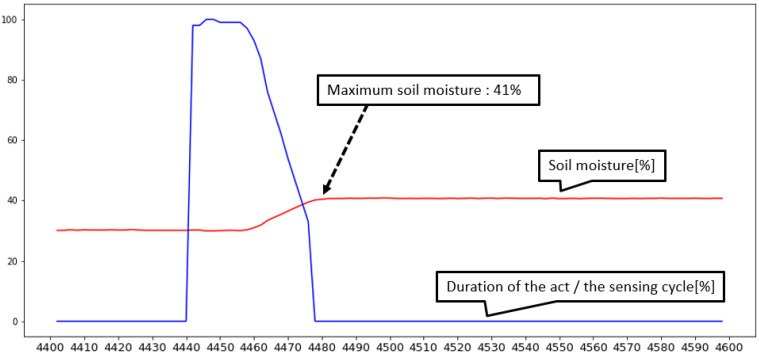
Sensed soil moisture values reflecting actuator actions in edge-level closed-loop control.

**Figure 21 sensors-22-03843-f021:**
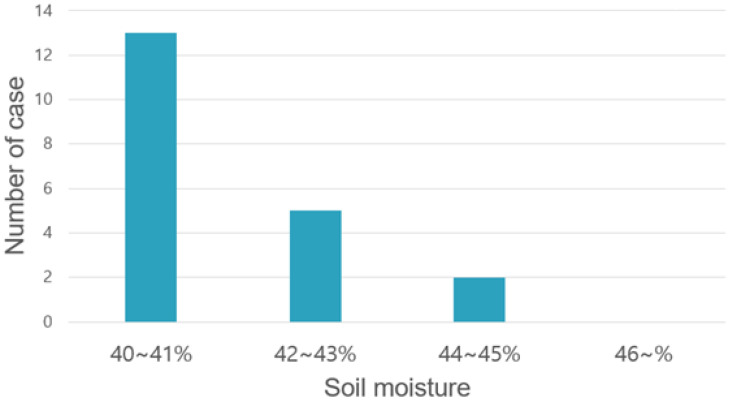
Case distribution depending on the maximum soil moisture in edge-level closed-loop control.

**Figure 22 sensors-22-03843-f022:**
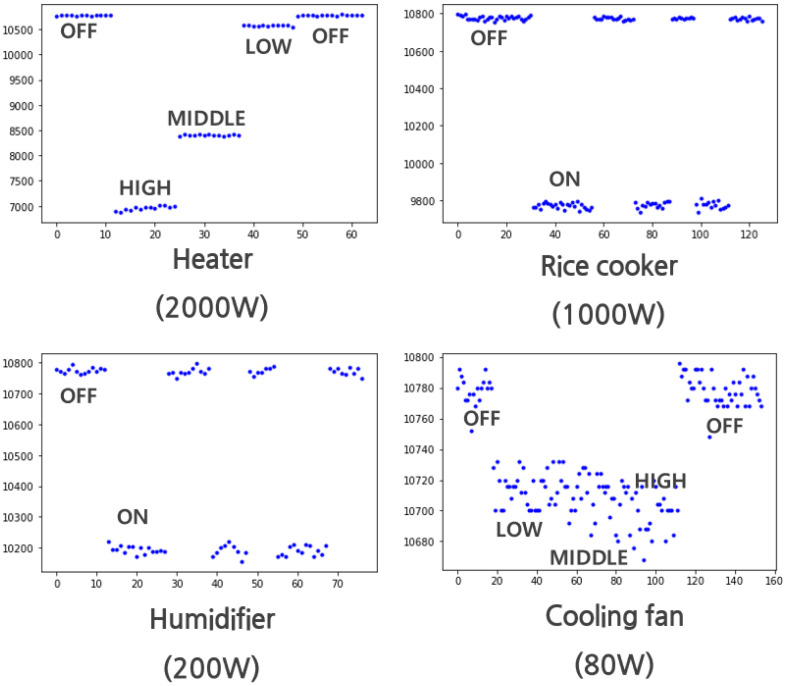
Current amplitude status according to activated appliances.

**Figure 23 sensors-22-03843-f023:**
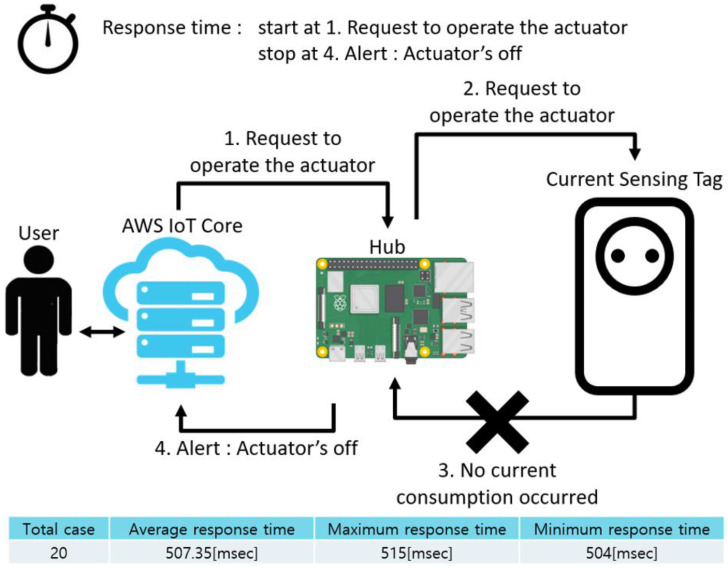
Response time in exceptional case: actuator off.

**Figure 24 sensors-22-03843-f024:**
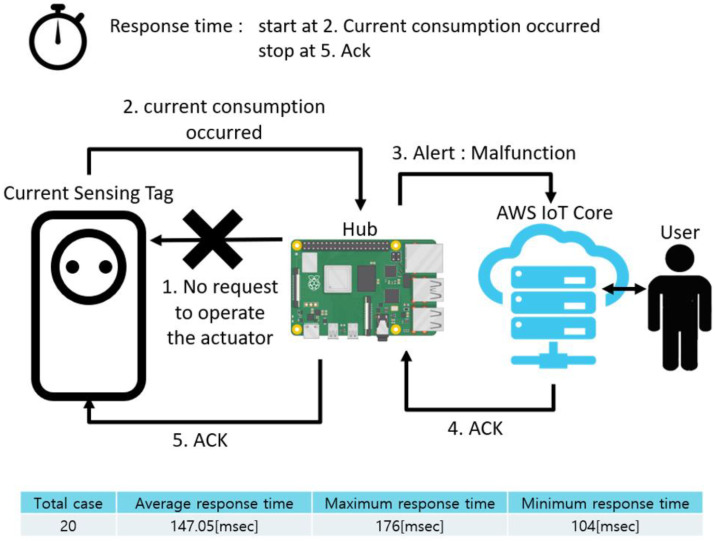
Response time in exceptional case: malfunction.

**Figure 25 sensors-22-03843-f025:**
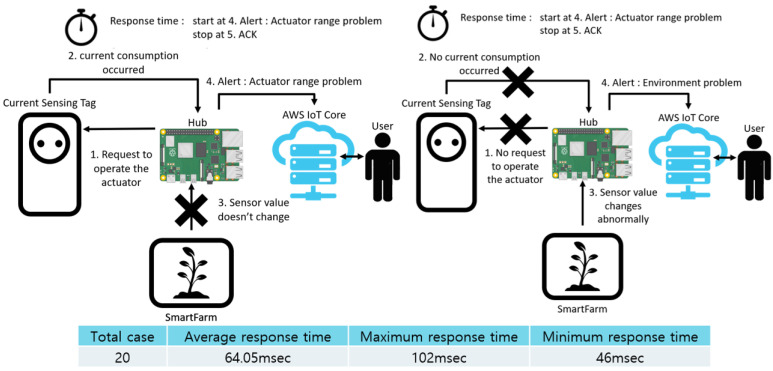
Response time in exceptional case: actuator range problem and environment problem.

**Table 1 sensors-22-03843-t001:** Table showing alert cases.

Situations	Request of the Actuator Operation	Power Consumption of the Actuator is Detected	Environment Sensor Value is Changed
Actuator off	True	False	-
Malfunctioning	False	True	-
Actuator range problem	True	True	False
Environment Problem	False	False	True

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
