# Peer review of "Secured and Deterministic Closed-Loop IoT System Architecture for Sensor and Actuator Networks"

_sensors, 2022, doi:10.3390/s22103843_

Round 1
Reviewer 1 Report
Introduction can be improved with much more facts from other studies not only in 2 sentences about the actual stadium in this domain.
It will be good to connect the analytical model with the experiences date measurement in the actual study.
It can be seen the research is very large but the structure of the presentation can be optimized for a easy understanding.
The conclusion can be improved with the percent and date present already in the research.
Author Response
Journal : Sensors (ISSN 1424-8220)
Manuscript ID : sensors-1679316
Title :Secured and Deterministic Closed-loop IoT System Architecture for Sensor and Actuator Networks
Authors : Hyeon Su Kim , Yu Jin Park , Soon Ju Kang *
Response to Reviewers:
Dear Reviewers,
Thank you for your kind and detailed review.
The reviewing issues point out were fixed as follows:
--------------
- answer: As you pointed out, article has been improved by adding additional reference data and analysis of the conclusion of the current paper. Testing of the proposed system is still in progress in various fields, and we plan to proceed in connection with the real environment through more numerical and statistical models through future research. Related works have been added as follows.
Various IoT technologies, communication methods, and decision-making algorithms are used in the smart farm field. H.Navarro-Hellín et al. [8] uses a wireless sensor node for the irrigation water management system, and transmits sensor data to the server through the cellular network of GSM/GPRs. In order to implement a large-scale smart farm, a highly scalable wireless communication network is required, and a cellular network with a wide operating range is mainly used. In this paper, we propose a platform that controls a group of sensor nodes within a smaller area and adjusting water supply. In the pro-posed platform, edge level close-loop control is performed, however, there is a plan to use the existing WiFi or Cellulor network to store sensor data to the server through the gate-way device in future works.
Viani et al. propose Fuzzy-based decision support system for irrigation management in vineyard .[9] In the proposed system, they designed a wireless sensor and actuator network (WSAN) architecture to collect data and perform irrigation control. The actuator on-off was scheduled through the collected data and a decision support system based on fuzzy logic, effectively reducing water consumption for crops. Control platform in the smart farm must consider various environmental variables and control factors such as moisture control, nutrient supply, disease prevention.[10] However, in this paper, only the method of maintaining the soil with constant humidity through moisture control is con-sidered.
In recent years, research has been actively conducted on the use of machine learning to make decisions for growing crops.[11] We are also considering applying server or Ti-nyML-based edge-level machine learning technology to control based on collected sensor data in next study. Short-term moisture control requires an immediate response, so it is advantageous to handle it within the WSN network. However, in the case of long-term moisture control, various factors need to be considered, so we plan to study a machine learning-based control platform by accumulating sensor data and result values on the server.
Reviewer 2 Report
I have read "Secured and Deterministic Closed-loop IoT System Architecture for Sensor and Actuator Networks" by Hyeon Su Kim et al. Below you can find my view of some issues that I have found in the paper.
Line 35: Authors say that determinism is "reduced". I doubt that determinism can be measured as a continuous value. Can something be half-deterministic? So is it or it is not deterministic? Please drop this approach and wording as it does not hold. The approach presented here seems to reside well in fuzzy logic and is far from hard real-time. Discussion of system behavior provided in lines 279-288 confirms my view.
Lines 40-69: While analyzing the architecture I find it quite strange that status of device is assessed by remote measurement of energy consumption. If the device has enough communication capabilities to receive remote commands it should be ready to measure its own energy consumption. Using wall socket seems very strange. Please do not take me wrong - I understand usage of current sensor in AC socket. It is for home or office premises owner who would like to know power consumption of devices to possibly replace equipment and optimize payments. In case if need of control over power-line device a simple relay board with IoT connectivity features would be enough.
Furthermore measurement of current to the device tells little about the status of the device. It may malfunction but still consume the same amount of energy. Self-diagnostics should be available on the device itself as it has BLE connectivity.
Line 55: BLE abbreviation is used in the text and I assume it refers to Bluetooth Low Energy but it may confuse readers and hard to analyze in years to come. Please expand abbreviations when first used.
Line 78: Kunert article is about reliability of network throughput and not about the safety. Nowadays sniffing messages in system that employs FHSS is trivial. FHSS improves multichannel usage of given band if wideband communication is not available. However, nowadays it is rather a liability due to channel switch overhead and synchronization requirements.
Line 86: Authors mention TDMA/CDMA/FDMA usage in MAC layer of IEEE 802.15.4. This is the first time that I see CDMA supposedly to be available in MAC layer of IEEE 802.15.4. Could you please provide reference for that information or verify and fix the text?
Line 104: again Authors claim that abnormal situation can be detected on the basis of current consumption. It would mean that all kinds of device abnormal behavior would change current consumption. I think that device may perform incorrect operations having the same current level. Measurement of AC in case of small device powered via impulse power supply is barely feasible. If the current measurement board is suitable for typical current range of AC socket (so up to 16 A) then change in power consumption would be just a few bits in case of 12-bit ADC. This discussion is all very vague if the connected device is just a generic "device". So, could you please provide boundary conditions for the "device" and current measurements? What currents did you observe? Figure 16 has no units.
Lines 119-123: why to control AC socket when one can easily control electronic device with communication capability? Why to acquire one AC socket to control one device when several could be connected to it via electric distributor? Why work on high voltage side while it is fairly easy to change device power consumption by sleep methods?
Line 135-136: when home/office communication to the cloud is employed via the Internet then there is no determinism. In normal circumstances ISP networks are not deterministic to end users. Deterministic network is possible but far beyond financial capabilities of most users. So let us assume it is not "difficult" but simply impossible.
Line 184: brackets
Lines 233-255: could you please be more specific about the hardware board? Was it designed by the authors or is it some COTS solution? What MCUs are in use there?
From the paper I do not know how specific AC socket with CST is recognized and addressed. Could you please explain it?
Lines 383-394 and figure 19: Authors claim that 40% soil humidity is "deterministic" and above this value up to 46% it becomes "Not deterministic". However, what happens below 40% humidity? Is it ultradeterministic? And what about values above 46% - are they less than "not deterministic" thus chaotic? I know deterministic soil humidity equal to 100% - that is the bottom of the river. Similarly deterministic is desert sand that has very low humidity. Figure 19 and its discussion has to be rewritten and redrawn as it is meaningless now. Perhaps authors tried to implement PID control loop and describe stability analysis without proper wording?
Line 430: Was the measurement resolution really equal to 20 Amperes? Even 5A is huge even in terms of power consumption as it provides over 1kW of power. If it is resolution then in typical power-line socket (16A-25A) it would give about 2-bit resolution of measurement. So what are maximum currents possible to be measured in the power line which is used by the Authors? Could you please verify that measurement resolution is actually 20 A or even 5 A?
Line 442: this figure and few following ones confirm that term "deterministic" cannot be applied to presented system as an "average" value is provided with both "minimum" [observed] and "maximum" [observed] values. This is probabilistic approach, reactive approach with undefined response time.
Summing up, this research does not solve an issue of deterministic control loop as it uses non-deterministic approach and methods. Please reconsider what "determinism" is and why it obviously does not apply here. Term "safety" also appears occasionally but is not discussed well. Hard to tell in which area the "safety" is Authors' concern. On the other hand the title claims "secured" but there is nothing about security in the text. Therefore a change of the title is a necessary first step to rewrite the paper.
My review may sound harsh but I consider the study of the created system as promising and this is first draft of the article. It has some strong points such as quite well designed architecture and practical usability. I suggest to emphasize these strong points and do not claim more than actually was provided in the hardware and software.
Author Response
Journal : Sensors (ISSN 1424-8220)
Manuscript ID : sensors-1679316
Title :Secured and Deterministic Closed-loop IoT System Architecture for Sensor and Actuator Networks
Authors : Hyeon Su Kim , Yu Jin Park , Soon Ju Kang *
Response to Reviewers:
Dear Reviewers,
Thank you for your kind and detailed review.
The reviewing issues point out were fixed as follows:
Line 35:
Answer) The approach in this paper is to consider the system requirements and real-time performance required in the actual application environment. The method using the server is difficult to satisfy system requirements such as response time because there is communication overhead and uncertainty of communication. For this reason, in this paper, we propose a system that can satisfy system requirements by reducing the response time by making judgment and actual control at the edge level. In addition, in this study, it was judged that the response at the edge level is advantageous because it is easy to guarantee a accurate and fast communication speed through a resource such as wired communication even in an area where determinism is required in the system. However, as you pointed out, we do not think it is correct to use the expression to reduce or guarantee determinism directly. Accordingly, the text has been fixed as follows.
Line 35) This overhead reduces the determinism of the application.
-> This overhead makes it difficult for the system to satisfy the system requirements and furthermore to guarantee determinism.
Lines 40-69:
Answer) In this paper, by separating the power measuring device, we tried to design it in a way that helps the system malfunction or change due to external interference. In our previous study, such a method of measuring power and reporting malfunctions was performed with one device. However, we thought that separating the devices has advantages such as security issues and process distribution that may occur in real environments. Even if the FEU is stopped, it can be discovered that the actuator operates by external intervention by the separated CST.
The classification of malfunctions is covered in chapter 3.4. Also, the device operate detection method is described in “Chapter 4.2.3 Classification of application operates”. In this paper, we aim to detect action that does not match the control signal. And we also tries to detect action by external manipulation. CST recognizes device operation through amplitude analysis of current signal and device calibration.
Line 55:
Answer) We have fixed the text. -> Bluetooth Low Energy(BLE)
Line 78 & 86:
Answer) As you pointed out, the related research part was revised because it was thought that the related research was far from the current topic. We tried to describe that various communication methods are used in the IoT device-based smart farm technology research, but we think the selection and technology method of the related research is wrong. The related research has been rewrote as follows :
Various IoT technologies, communication methods, and decision-making algorithms are used in the smart farm field. H.Navarro-Hellín et al. [8] uses a wireless sensor node for the irrigation water management system, and transmits sensor data to the server through the cellular network of GSM/GPRs. In order to implement a large-scale smart farm, a highly scalable wireless communication network is required, and a cellular network with a wide operating range is mainly used. In this paper, we propose a platform that controls a group of sensor nodes within a smaller area and adjusting water supply. In the proposed platform, edge level close-loop control is performed, however, there is a plan to use the existing WiFi or Cellulor network to store sensor data to the server through the gateway device in future works.
Viani et al. propose Fuzzy-based decision support system for irrigation management in vineyard .[9] In the proposed system, they designed a wireless sensor and actuator network (WSAN) architecture to collect data and perform irrigation control. The actuator on-off was scheduled through the collected data and a decision support system based on fuzzy logic, effectively reducing water consumption for crops. Control platform in the smart farm must consider various environmental variables and control factors such as moisture control, nutrient supply, disease prevention.[10] However, in this paper, only the method of maintaining the soil with constant humidity through moisture control is considered.
In recent years, research has been actively conducted on the use of machine learning to make decisions for growing crops.[11] We are also considering applying server or TinyML-based edge-level machine learning technology to control based on collected sensor data in next study. Short-term moisture control requires an immediate response, so it is advantageous to handle it within the WSN network. However, in the case of long-term moisture control, various factors need to be considered, so we plan to study a machine learning-based control platform by accumulating sensor data and result values on the server.
Line 104:
Answer) As mentioned above, a test graph of the current measured through the ACS-712 is included in the paper. CST classifies the operating state by converting the amplitude of the current value, and separates On/Off and Low/High state. Here, the proposed system detects an error condition by checking the equality of the measured state value and the current control command. In order to increase the accuracy of this technology, we are currently studying a method of calculating the current value by mixing the integration data of the current value.
Lines 119-123:
Answer) The CST used in this study was developed for general-purpose research and development. The CST can be utilized not only in the current smart farm, but also in various fields such as Home Appliance and Smart Factory through sensor replacement. For scalability and research and development time, the developed CST board was directly connected to a commercial AC socket and used. Depending on the actual application environment of the device, it is also considered a good way to implement it with multiple sockets. In the proposed system, independent power control is required to stop the device in an uncontrollable state.
Line 135-136 :
Answer) This platform does not aim to realize determinism through ISP Network. The proposed edge-level system aims to gain benefits in fast responsiveness, control, and security compared to using a server. However, we think there were many expressions that contradicted the point in the paper. Therefore, we plan to change the relevant part of the current paper and conduct research to apply the part related to Determinism to the proposed system in future research. We have corrected the expression of that part in the current paper.
Line 184 :
Answer) We have fixed the text.
Lines 233-255:
Answer) All hardware is custom developed except for commercial AC sockets. The AC socket only serves to provide a device for current measurement. The proposed CST board was developed using nRF52840 (cortex-m4) MCU, and it performs Bluetooth communication and current signal processing. The CST board measures the current by connecting the power supply of the AC socket to the ACS sensor in series.
Lines 383-394:
Answer) The test of the proposed system uses 40% as a reference value to confirm how much actual control errors are caused by the difference in response time between the cloud server and edge level control. As you said, the graph expressed in determinism was modified because it was not appropriate.
Lines 430:
Answer) This part is misrepresented. ACS has the “operating range” of 0~5A / 0~20A / 0~40A. The ACS board uses the nRF52840 MCU's 14-bit resolution ADC to measure current values. In Korea, where our research was conducted, AC 220V is generally used. Because of this, home appliances with power consumption lower than 1100W use the ACS712(5A). The CST board can be used interchangeably with 5/20/40A versions.
Lines 442:
Answer) We thought that determinism in the conclusion part was not an appropriate expression, so we modified it. Considering determinism in our current platform is a necessary part, so we plan to address these issues through future research.

Reviewer 3 Report
Dear Authors
Your research is very interesting, you propose a sensor architecture, which allows autonomous, local decision making. This is very good because it solves the problem of latency or loss of connection between the sensors and the Internet (cloud services).
- Your paper provides very clear information, which allows us to see that the stated objective has been successfully achieved.
- I suggest that you organize the content in a better way, in some sections, there is a lot of text, piled up.
- Edit the recommendations, and make them more concrete.
Author Response
Journal : Sensors (ISSN 1424-8220)
Manuscript ID : sensors-1679316
Title :Secured and Deterministic Closed-loop IoT System Architecture for Sensor and Actuator Networks
Authors : Hyeon Su Kim , Yu Jin Park , Soon Ju Kang *
Response to Reviewers:
Dear Reviewers,
Thank you for your kind and detailed review.
The reviewing issues point out were fixed as follows:
--------------
- answer: Relevant research has been added to the entire article, and parts that are not smooth have been reinterpreted and reconstructed. In addition, problems were found and corrected in the conclusion and test contents.
Reviewer 4 Report
- What is the main question addressed by the research?
The main goal of this research seems to be the capacity for improvement referred to as closed-loop systems in IoT environments that rely on cloud-level controlling schemes, applicable to farming and industrial fields. The authors strive to reduce communication overhead with the cloud whilst placing computing nodes, serving as gateways between the cloud and the edge sensing nodes, and implementing MQTT communication services.
Do you consider the topic original or relevant to the field? Does it
address a specific gap in the field?
I consider the topic as relevant to the field, as the authors introduce a new method to improve environmental control while remotely determining the actuator's behavior from the cloud whilst being placed on the field of the application, making use of sensor values at the edge level.
It does address a specific gap that is the determination of safety measures to control an actuator remotely from the cloud and simultaneously make use of sensor signals on the field keeping data flow as restricted in magnitude as possible.
- What does it add to the subject area compared with other published material?
They propose an improved system remote-controlled environment system architecture that monitors plant conditions for irrigation and fertilization depending on the soil and plant conditions which is a representative example of closed-loop IoT networks. This stands for a significant issue globally having to do with efficient farming.
The authors uniquely propose the use of a current-sensing tag to monitor the actuator's operation requests, and their current consumption according to a variety of loads, to monitor and inform the users away from the field, making use of MQTT messages in case of any system malfunctions.
Although similar computation nodes have already been presented as solutions for low impact in cloud implementations, according to their approach the use of such a system enhances the safety of a remote control system by distinguishing whether the actuator is being operated under a normal system command or a non-system factor.
They addressed message time delays on an experimental basis giving a magnitude of expectancies in the time domain.
Their experimental confirmation via simulation addresses four abnormal realistic situations that may occur during remote control, where the proposed architecture provides MQTT warning messages accordingly while occurring.
The current sensor implementation proves to be a standalone proposal.
- What specific improvements should the authors consider regarding the methodology? What further controls should be considered?
Session 2 should be enhanced in references to relevant works regarding wireless communication safety measures, relevant remote control applications, computational nodes, and suggested computation platforms.
The proposal could benefit from a greater variety of AC loads, a more precise AC sensor, and different computational platforms for merely comparison reasons, to provide an even more accurate presentation of everyday farming challenges.
Improvements in text:
- Table 1. page 7: “ X” and “O” situations should be specified to match the operating status stated in the following corresponding text i.e text lines 224 - 232. Words such as operation, yes, and true may be more suitable, as well as a subtext clarifying all conditions under Table 1.
- Line 245: Should the word Cortex be added before “M0+”
- Are the conclusions consistent with the evidence and arguments presented and do they address the main question posed?
The conclusions seem to be consistent with the presented evidence and they address the main subject adequately.
First of all the authors address an existing problem having to do with remote-controlled environment systems that monitor plant conditions to further introduce their methodology by providing clarifications on required terminology.
They refer to some related works about safety problems associated with remote control, in industrial fields and wireless closed-loop system application cases.
Then they propose and evaluate the use of a dedicated BLE communication board in combination with a relay hub, tested in use with a variety of AC loads in various conditions. A custom-made Current -sensing tag is put to test, to monitor actuator operation requests, and the current consumption of actuators to determine and inform the user of abnormal situations. All evaluated performance conditions such as AC bias current, are explicitly handled.
They analyze the performance of the proposed communication implementation regarding errors in remote actuator control situations applicable in realistic events on the field.
On the whole, their approach may improve closed-loop systems in IoT environments that rely on cloud level controlling schemes, applicable to farming and industrial fields.
- Are the references appropriate?
On the whole, the authors have used appropriate references to:
- describe the closed-loop performance at the cloud and edge levels,
- to compare their proposal referring to a remotely-controlled environment system that remotely monitors plant conditions and supplies water or liquid fertilizers depending on the conditions, claiming to be a representative example of a closed-loop IoT network,
- and refer to related work such as safety problems associated with remote controlling in farming establishments.
Author Response
Journal : Sensors (ISSN 1424-8220)
Manuscript ID : sensors-1679316
Title :Secured and Deterministic Closed-loop IoT System Architecture for Sensor and Actuator Networks
Authors : Hyeon Su Kim , Yu Jin Park , Soon Ju Kang *
Response to Reviewers:
Dear Reviewers,
Thank you for your kind and detailed review.
The reviewing issues point out were fixed as follows:
--------------
- answer: We have corrected the recommendations you pointed out. In addition, the expression throughout the article has been corrected and related research has been added to further explain the proposed platform.
Round 2
Reviewer 2 Report
I generally accept the rebuttal letter and appreciate modifications made to the paper. This is solid work of research and engineering despite far claim over "determinism".
I believe that in fig 19 and 21 there should be "Number of cases" (plural) instead of "case" (singular).
Please use spellchecker before providing final text.